# Outcome of Photoselective Vaporization of the Prostate with the GreenLight-XPS 180 Watt System Compared to Transurethral Resection of the Prostate

**DOI:** 10.3390/jcm8071004

**Published:** 2019-07-10

**Authors:** Maximilian Reimann, Nikita Fishman, Isabel Lichy, Laura Wiemer, Sebastian Hofbauer, Zenai Almedom, John Buckendahl, Ursula Steiner, Thorsten Schlomm, Frank Friedersdorff, Hannes Cash

**Affiliations:** Department of Urology, Charité–University Medicine Berlin, 12203 Berlin, Germany

**Keywords:** greenlight, TURP, PVP, photoselective vaporization, BPE, prostate

## Abstract

The aim of this paper was to compare the perioperative and postoperative results of photoselective vaporization of the prostate with the GreenLight-XPS 180 Watt System (PVP) and transurethral resection of the prostate (TURP). This retrospective study included 140 men who underwent PVP and 114 men who underwent TURP for symptomatic benign prostate enlargement (BPE) between June 2010 and February 2015. The primary outcome measures were the patient reported outcome, operative results, International Prostate Symptom Score-Quality of Life (IPSS-QoL), complication rates, catheterization time, and length of hospital stay. The median follow-up times were 27 months (range 14–44) for the PVP group and 36 months (range 25–47) for the TURP group. The patient characteristics were well balanced in both groups with a median age of 71 years (PVP group) vs. 70 years (TURP group) and a comparable prostate volume (median 50 mL in the PVP group vs. 45 mL in the TURP group). The IPSS-QoL was significantly higher in the PVP group than in the TURP group (median 22 + 4; range 16–27 + 3−5 vs. median 19 + 3; range 15−23 + 3−4; *p* = 0.02). Men undergoing PVP were more likely to be on anticoagulants (PVP group *n* = 23; 16% vs. TURP group *n* = 2; 2%, *p* < 0.001). The median operation time (OT; min) for both procedures was comparable with 68 min (PVP group; range 53–91) vs. 67 min (TURP group; range 46–85). The rate of severe intraoperative bleeding was significantly lower in the PVP group than in the TURP group (*n* = 7; 5% vs. *n* = 16; 14%; *p* = 0.01). The postoperative catheterization time and length of hospital stay was significantly lower in the PVP group (median 1–2 days; range 1–4) vs. the TURP group (median 2–4 days; range 2–5; both *p* < 0.001). Complication rates (Clavien-Dindo classification ≥III) based on the follow-up data showed no statistically significant difference between the PVP group and the TURP group (*n* = 6; 4% vs. *n* = 6; 5%; *p* = 0.28). The IPSS on follow-up showed an equivalent reduction in symptoms for both treatment modalities (IPSS-QoL of 5 + 1; range 2–11 + 0−2 for both). There were no differences concerning urge (PVP group *n* = 3; 2% vs. TURP group *n* = 3; 3%; *p* = 0.90) and men were similarly satisfied with the postoperative outcome (PVP group 92% vs. TURP group 87%; *p* = 0.43). The PVP group was associated with a shorter hospitalization time and showed a reduced risk of bleeding, despite patients remaining on anticoagulants, without increasing the overall operative time. There was no difference in the patient reported outcome for both procedures.

## 1. Introduction

The incidence of symptomatic benign prostate enlargement (BPE) increases in males in the later decades of life. BPE can be diagnosed in 50–60% of men in their sixth decade of life and up to 90% of men in their eighth decade of life [1,2,3]. The steady progressive nature of this disease comes naturally with failure in conservative medical treatment over the years. Patients with symptomatic BPE have persistent lower urinary tract symptoms (LUTS) and require surgical intervention for subvesical desobstruction. 

GreenLight-XPS 180 Watt photoselective vaporization of the prostate (PVP) and transurethral resection of the prostate (TURP) are both surgical options for the treatment of BPE [4]. TURP is currently considered the surgical gold standard in BPE treatment, but may be associated with several complications such as perioperative bleeding [5]. Alternatives that provide a lower risk of bleeding are required, especially for older patients with cardiovascular diseases that are in need of a permanent anticoagulation treatment. PVP was added as one endourologic method to the European guidelines in 2010, and as a therapeutic recommendation to the American guidelines in 2018 [4,6,7]. The advantage of this method is that the 532 nm wavelength visible green light that is used is strongly absorbed by oxyhemoglobin and provides simultaneous vaporization and coagulation of the prostatic tissue [8].

The Goliath study confirmed the non-inferiority of PVP compared to TURP in regard to the International Prostate Symptom Score (IPSS), Qmax, and complication rates in 2012 [9,10,11]. PVP with 180 watts has been available since 2010, although other systems with lower energy were available prior to this. Since then, the way PVP is applied may have improved with growing experience. However, there are currently few evaluations of the use of PVP in clinical settings. We were able to show an improvement in the perioperative parameters of patients who underwent PVP within a timeframe of five years [12]. The primary intention of this study was to evaluate the results and patient-related outcomes of PVP and TURP.

## 2. Patients and Methods

### 2.1. Patients and Study Design

Institutional ethics committee approval was granted. Informed consent was obtained from all individual participants included in the study. The retrospective single center analysis included 375 men who underwent PVP (Boston Scientific, Minnetonka, MN, USA) and 443 men who underwent TURP for symptomatic BPE between June 2010 and February 2015 at Charité–University Medicine Berlin. In total, we included *n* = 140 (PVP group) and *n* = 114 (TURP group) patients who participated in the postoperative follow-up for this study.

The primary outcome measurements were operation time (OT; min) and laser time (LT; min) separated in patient groups based on prostate volume (PV; mL) (group 1 < 40 mL up to group 4 > 80 mL; 20 mL steps) and year of surgery (2010–2015) in respect to effectiveness, efficacy, and safety, with a postoperative follow-up. A prolonged hospital stay was defined as >2 days for PVP and >4 days for TURP.

Postoperative follow-up included International Prostate Symptom Score-Quality of Life (IPS-QoL) and adverse events (AEs; intraoperative and postoperative as stated by the patients). The timepoint of the adverse events were categorized into early (<30 days), mid (30–180 days), and late (>180 days) events according to the time of occurrence (as documented by the patients). AEs were subcategorized based on grade (low-grade (Clavien-Dindo grade I–II) and high-grade (need of an intervention; Clavien-Dindo grade ≥IIIa–IV)), prostate-specific antigen (PSA; ng/mL), hospital stay (days) and catheterization time (days), reoperation, and postoperative re-catheterization.

### 2.2. Surgical Procedures

All main PVP and TURP surgeons were fully educated urologists equally experienced in performing PVP and TURP regularly. PVP was performed using a continuous-flow dedicated Storz laser 24 Charrière (Ch.) endoscope. All PVP patients were treated with a vaporization technique without any tissue resection. TURP was performed using a passive flow dedicated Storz 24 Ch. resectoscope with or without trocar cystostomy. During PVP, a suprapubic catheter (SPC) was placed in men with preoperative post residual volume based on the surgeon’s indication. After TURP, an SPC was placed over the suprapubic trocar cystostomy based on prostate volume and the surgeon’s preference. In all patients a transurethral three-way irrigation catheter was placed at the end of the procedure.

### 2.3. Statistical Analysis

We used IBM-SPSS (IBM, New York, NY, USA) for all statistical analyses. Statistical significance was assessed at the 5% level (two-sided; *p* < 0.05) using the two-sample *t*-test, Mann–Whitney U test, or for nominal data the chi-square test.

## 3. Results

Patient baseline characteristics of the 254 men (PVP group *n* = 140; TURP group *n* = 114) are summarized in Table 1. Men undergoing PVP were more likely to be on anti-obstructive medication (α-blocker; 5α-reductase inhibitors) (*n* = 109; 78%) compared to the TURP collective (*n* = 73; 64%; *p* = 0.02). Likewise, the IPSS-QoL was significantly higher in the PVP group (median 22 + 4; range 16.8−27 + 3−5) compared to the TURP group (median 19 + 3; range 15–23 + 3−4; *p* = 0.02). There were no statistical differences in the preoperative catheterization in the PVP collective (*n* = 46; 33%) compared to the TURP collective (*n* = 40; 35%; *p* = 0.71), or in the preoperative post-void residual urine volume (PVP group median 60 mL; range 0–138 mL vs. TURP group median 65 mL; range 0–175 mL; *p* = 0.78). Neither group differed in the rate of urinary tract infections on admission (PVP group 12% vs. TURP group 16%; *p* = 0.40).

Sixty-four patients in the PVP group were on acetylsalicylic acid, 23 (16%) of which remained on anticoagulation because of severe co-existing medical conditions compared to two men (2%) in the TURP group (*p* < 0.001). Clopidogrel in combination with acetylsalicylic acid was needed in five patients in the PVP group (4%) compared to three patients in the TURP group (3%), with a total of four patients in the PVP group (3%) and two patients in the TURP group (2%; *p* = 0.57) remaining on double anticoagulation. Phenprocoumon and new oral anticoagulants (NOACs) were bridged with therapeutic low molecular weight heparin in 14 patients (10%) with two patients (1%) remaining on anticoagulation in the PVP group compared to 10 patients (9%; *p* = 0.74) with one patient (1%; *p* = 0.69) in the TURP group as shown in Table 1.

The number of patients who were given preoperative and perioperative antibiotic prophylaxis was significantly higher in the PVP group with *n* = 116 (83%) compared to the TURP group with *n* = 31 (32%; *p* < 0.001), and this was given for a longer period with a median of 5 days in the PVP group compared to the TURP group with a median of 3 days (*p* < 0.02).

Intraoperative baseline results showed a similar median operation time (OT) for both procedures with no statistically difference in the PVP group with a median of 68 min (range 53–91) compared to the TURP group with a median of 67 min (range 46–85; *p* = 0.18). The effective OT per mL of prostate volume (PV) stayed stable in the PVP group with a median of 1.30 min/mL PV (range 1.12–1.8) compared to the TURP group with a median of 1.32 min/mL PV (range 1.0–1.8; *p* = 0.44), as shown in Table 2.

Intraoperative bleeding with the need for extensive coagulation (as stated in the surgeon’s surgical report) as the most common adverse event was significantly lower in the PVP group with a median of *n* = 7 (5%) than in the TURP group with a median of *n* = 16 (14%; *p* < 0.01).

Suprapubic catheter (SPC) placement was significantly lower in the PVP group (*n* = 27; 19%) compared to the TURP group (*n* = 64; 58%; *p* < 0.001), but was also associated with the routine placement of a suprapubic trocar cystostomy in the TURP group. The same applied to the rate of intraoperative bleeding (PVP group *n* = 7; 5% vs. TURP group *n* = 16; 14%; *p* = 0.01) as shown in Table 2.

Postoperative baseline results showed that the hospital stay was significantly lower in the PVP group with a median of 2 days (range 2–4) than in the TURP group with a median of 4 days (range 3–5; *p* < 0.001). Prolonged postoperative hospital stays were statistically significantly reduced and shorter in the PVP group with a median length of >2 days (*n* = 51; 37%) compared to the TURP group with a median length of >3 days (*n* = 66; 58%; *p* < 0.001), as summarized in Table 2.

Therefore, the catheterization time was also significantly lower in the PVP group with a median of 1 day (range 1–2) compared to a median of 2 days for the TURP group (range 2–3; *p* < 0.001). The postoperative SPC removal was significantly earlier after PVP with removal after a median of 2 days (range 2–3) vs. 3.5 days after TURP (range 3–4; *p* < 0.001), as shown in Table 2.

Overall adverse events, including long-term complication based on the follow-up analysis, showed no statistically significant differences between the PVP and TURP groups (*n* = 74; 53% vs. *n* = 68; 60%, respectively; *p* = 0.28). The detailed adverse events are shown in Table 3.

Postoperative re-intervention due to bleeding (Clavien-Dindo >IIIa) was necessary in the TURP group with a median of *n* = 3 (12%), compared to the PVP group in which no patients required a re-intervention because of bleeding (*p* = 0.09). Postoperative acute urine retention was the most common adverse event, which occurred equally in both groups (PVP group median *n* = 8; 6% vs. TURP group median *n* = 11; 10%; *p* = 0.24). The rate of postoperative urge incontinence on follow-up was comparable in both groups (PVP group *n* = 3; 2% vs. TURP group *n* = 3; 3%; *p* = 0.80), and the same applied for the need for re-intervention (PVP group *n* = 6; 4% vs. TURP group *n* = 11; 10% *p* = 0.09).

At a median follow-up of 27 months for the PVP group and 36 months for the TURP group, both groups showed comparable symptom reduction assessed by IPSS-QoL (PVP group median 5 + 1; range 3−10 + 0−2 vs. TURP group median 5 + 1; range 2−11 + 0−2; *p* = 0.64 + *p* = 0.49). When asked for treatment satisfaction and future treatment recommendations (scale from most satisfied, satisfied, unsatisfied, to most unsatisfied), both treatment groups were equally satisfied with the results of the intervention (at most satisfied/satisfied; PVP group *n* = 120; 92% vs. TURP group *n* = 84; 87%; *p* = 0.43).

## 4. Discussion

Both PVP and TURP present surgical options as recommended by current urological guidelines for subvesical desobstruction [4,7,10].

The present analysis sought to evaluate the results of PVP compared to TURP in a clinical setting. The overall OT is in line with data from the Goliath study with a median OT of 46 min (range 15–160) in the PVP group vs. a median of 36 min (range 0–160) in the TURP group [10]. The correlation of the OT to the PV with a longer surgical time for larger glands (PV >80 mL) was also stated by Meskawi et al., who showed a median OT of 90 min for men with a median PV of 120 mL [13]. Stone et al. published similar results with a median OT of 180 min in men with a median PV of 202 mL [14].

Recently published data by Valdivieso et al. underline that PVP presents a feasible treatment option for men with glands larger than 200 mL. In their analysis the median OT was 129 min and the authors found no difference in terms of functional outcome or complication rates [15]. Elshal et al. furthermore confirmed that PVP was non-inferior to Holium laser enucleation of the prostate (HoLep) as a size-independent technique to treat LUTS [16]. Zhou et al. also defined PVP as a size-independent technique with reproducible outcome when performed by an experienced surgeon applying an energy density of >4 kJ/mL PV, and PSA reduction at 6 months postoperatively occurred in >50% percent of the cases [17]. Our energy density is in line with these results (3.5 to 4 kJ/mL PV), which is also reflected in the reported patient outcome [12].

In our analysis a significant number of men treated with PVP remained on anticoagulation, which is in line with a comment by Rieken et al. who confirmed that PVP as a safer method in LUTS treatment, especially in high-risk patients, compared to TURP [18].

Overall, our analysis clearly shows the clinical equality for both techniques, which is of relevance since PVP with 180 watts was introduced in the year 2010 and clearly differs from the previous Greenlight systems. The reduction of OT and LT over time emphasizes the effectiveness and efficacy of PVP as well as the consistency of the method compared to TURP [12]. Teng et al. confirmed in their meta-analysis PVP as an alternative minimal invasive method for symptomatic BPE with similar results on subjective (IPSS-QoL) and objective (Qmax, PVR) outcome measurements [19].

Furthermore, the safety of PVP compared to TURP was highly confirmed in our study as the PVP group had a stable low complication rate comparable to the TURP group, despite the TURP group having a lower rate of existing co-morbidities and no permanent anticoagulation. Despite remaining on anticoagulation while performing PVP, there was no impact on morbidity or adverse events (need of an intervention; Clavien-Dindo >IIIa–IV) in the PVP group compared to the TURP group. The data published by Lee et al. confirm our results for patients staying on permanent anticoagulation with no impact on the postoperative complication rates, hospital stay length, OT, LT, or catheterization time [20].

Likewise, Lee et al. and Bachmann et al. confirmed that a conversion to TURP was size dependent (large prostates >80 mL), while permanent anticoagulation was not associated with a higher complication rate compared to men with large glands without permanent anticoagulation [11,20]. Cindolo et al. confirmed in a recently published study that long-term outcomes for Greenlight laser enucleation of the prostate (GreenLEP), anatomical PVP (aPVP), and standard PVP (sPVP) were comparable. However, GreenLep was associated with a shorter LT and less energy use despite significantly larger glands than in aPVP or sPVP, with all three techniques providing sufficient patient satisfaction [21]. GreenLEP lead to a faster desobstruction and possibly lower costs, especially in larger prostates. The authors furthermore emphasized that an aPVP provided similar sufficient outcomes compared to sPVP [22]. Future analyses should focus on how recent changes and developments in the surgical techniques will influence patient outcome after Greenlight 180 W treatment.

Our analysis is limited due to its retrospective nature, the fact that not all men responded to our questionnaire, and the variation in time period and length. Additionally, the men in our study did not undergo clinical examination and therefore clinical parameters such as uroflowmetry and PSA reduction were unavailable.

## 5. Conclusions

PVP is as a safe and effective surgical method for symptomatic BPE and is comparable to TURP. Our single-center experience showed a dramatic improvement in the hospital stay length, a reduced risk of bleeding despite remaining on anticoagulation, and a comparable OT and effective LT whilst maintaining a stable low complication rate for PVP patients. Patient reported outcome was comparable for both interventions.

## Figures and Tables

**Table 1 jcm-08-01004-t001:** Patient baseline characteristics.

	PVP *n =* 140	TURP *n =* 114	*p* Value
Median age (years)	71 (65–75)	70 (66–75)	0.70
PSA (ng/dL)	3.1 (1.6–6.5)	2.6 (1.1–5.4)	0.36
Prostate volume (mL)	50 (35–69)	45 (34–70)	0.24
Median urinary retention (mL)	60 (0–138)	65 (0–175)	0.78
IPSS	22 (17–27)	19 (15–23)	**0.02**
QoL	4 (3–5)	3 (3–4)	0.10
Previous TURP	2 (1.4%)	11 (9.7%)	**0.03**
Previous PVP	1 (0.7%)	2 (1.8%)	0.46
Anticoagulation use of ASA	38 (27.1%)	26 (22.8%)	0.43
Continue ASA till OR	23 (16.4%)	2 (1.8%)	**<0.001**
Phenprocoumon/NOA	14 (10%)	10 (8.8%)	0.74
Continue Phenprocoumon/NOA till OR	2 (1.4%)	1 (0.9%)	0.69
Clopidogrel	5 (3.5%)	3 (2.6%)	0.67
Continue Clopidogrel till OR	4 (2.9%)	2 (1.8%)	0.57

PSA (prostate-specific antigen); IPSS (international prostate symptom score); QoL (quality of life); ASA (acetylsalicylic acid); NOA (new oral anticoagulant); OR (operation); GreenLight-XPS 180 Watt photoselective vaporization of the prostate (PVP); transurethral resection of the prostate (TURP).

**Table 2 jcm-08-01004-t002:** Intraoperative and postoperative baseline results.

	PVP *n =* 140	TURP *n =* 114	*p* Value
Median OT (min)	68 (53–91)	67.5 (46–85)	0.18
SPC placements (cases)	27 (19%)	64 (58%)	**<0.001**
Median OT/PV (min/mL)	1.30 (1.1–1.8)	1.32 (1.0–1.8)	0.44
Intraoperative bleeding (cases)	7 (5%)	16 (14%)	**0.01**
Median hospital stay (days)	2 (2–3)	4 (3–5)	**<0.001**
Prolonged hospital stays (cases)	51 (37%)	66 (58%)	**0.001**
Catheterization time (days)	1 (1–2)	2 (2–3)	**<0.001**
SPC removal (days)	2 (2–3)	3.5 (3–4)	**<0.001**

OT (operation time); SPC (suprapubic catheter); PV (prostate volume).

**Table 3 jcm-08-01004-t003:** Adverse events (early, mid-time, and late).

	PVP *n =* 140	TURP *n =* 114	*p* Value
**Early adverse events (<30 days; cases)**			
Acute urine retention	8 (5.7%)	11 (9.6%)	0.24
Prolonged hematuria	15 (10.7%)	14 (12.3%)	0.70
Infections	4 (2.9%)	4 (3.5%)	0.77
Urge incontinence	3 (2.1%)	3 (2.6%)	0.80
Dribbling incontinence	1 (0.7%)	3 (2.6%)	0.22
Bladder neck contracture	1 (0.7%)	1 (0.9%)	0.88
Urethra stricture	1 (0.7%)	0	0.37
Erectile dysfunction	2 (1.4%)	2 (1.8%)	0.84
Prostatitis	1 (0.7%)	0	0.37
Urosepsis	1 (0.7%)	0	0.37
Bladder cramps	1 (0.7%)	0	0.37
Fornix rupture	0	1 (0.9%)	0.27
Hypokalemia	1 (0.7%)	0	0.37
Fever	0	1 (0.9%)	0.27
Anemia	1 (0.7%)	1 (0.9%)	0.88
Unspecified pain	1 (0.7%)	1 (0.9%)	0.88
Diarrhea	2 (1.4%)	0	0.20
Heart rhythm disorder	1 (0.7%)	0	0.37
Pneumonia	1 (0.7%)	0	0.37
Unknown	1 (0.7%)	0	0.37
Total	46 (32.9%)	42 (36.8%)	0.51
**Mid-time adverse events (30–180 days)**			
Acute urine retention	2 (1.4%)	0	0.20
Urge incontinence	1 (0.7%)	0	0.37
Infection	1 (0.7%)	1 (0.9%)	0.88
Prolonged hematuria	2 (1.4%)	3 (2.6%)	0.49
Total	6 (4.3%)	4 (3.5%)	0.75
**Late adverse events (>180 days)**			
Acute urine retention	2 (1.4%)	0	0.20
Hematuria	1 (0.7%)	2 (1.8%)	0.46
Bladder neck contracture	0	1 (0.9%)	0.27
Total	3 (2.1%)	3 (2.6%)	0.80

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
