# Peer review of "Outcome of Photoselective Vaporization of the Prostate with the GreenLight-XPS 180 Watt System Compared to Transurethral Resection of the Prostate"

_jcm, 2019, doi:10.3390/jcm8071004_

Reviewer 1 Report

This is a well-presented study and a well-written manuscript. This is not an entirely novel study but the quality of the presentation is high and the cohort size large which makes this study interesting to urologists.

The authors state that they placed suprapubic tubes in 19% of PVP patients and 58% of TURP patients. This is an unusually high number and the authors may want to add what the indication for SPT placement was.

Author Response

Point by point reply

Reviewer: 1

This is a well-presented study and a well-written manuscript. This is not an entirely novel study but the quality of the presentation is high and the cohort size large which makes this study interesting to urologists.

The authors state that they placed suprapubic tubes in 19% of PVP patients and 58% of TURP patients. This is an unusually high number and the authors may want to add what the indication for SPT placement was.

Answer:

We thank the reviewer for his/her positive remarks.

We have added an explanation for the indication for SPT placement in the method section. The high SPT placement in the TUR-P group is associated with the placement of a suprapubic trocar cystostomy during the procedure in most men depending on their prostate volume and surgeon preference. We have added at statement in the results section line 131-133.

Reviewer 2 Report

Thank you for submitting a good manuscript which summarized a single center cases with surgical candidate of BPH. This study compared PVP(180Watt system) and TURP. Here are several questions and comments:

According to the title of your manuscript, “patient reported outcome”(PRO) would be the main focus of this study. But, not much of the descriptions are related to PRO. Rather, this manuscript described more about objective parameters. And to me, it seems that it focused more on the cases with anticoagulant uses. So I recommend to change something in your manuscript regarding this issue.

I want to make sure that the term “BPO” is properly used in the Introduction session. Please recheck whether the term “BPO” can be used without urodynamic study for each patient.

Line 122, What do you mean “intraoperative bleeding”? How did you define and measure?

In the Table 4, You included bladder neck contracture, erectile dysfunction & retrograde ejaculation in “early adverse events(<30 days: cases)”. But it is difficult to think those events are meaningful complications in the early postoperative period. Rather those should be measured and compared in the late stage. What is your thoughts?

In Conclusions session, the authors still summarizes several things rather than PRO itself. So it would be better to be revised little bit.

Many thanks.

Author Response

Comments and Suggestions for Authors

Thank you for submitting a good manuscript which summarized a single center cases with surgical candidate of BPH. This study compared PVP(180Watt system) and TURP. Here are several questions and comments:

According to the title of your manuscript, “patient reported outcome”(PRO) would be the main focus of this study. But, not much of the descriptions are related to PRO. Rather, this manuscript described more about objective parameters. And to me, it seems that it focused more on the cases with anticoagulant uses. So I recommend to change something in your manuscript regarding this issue.

Answer:

We thank the reviewer for his/her important remark. We have changed the manuscripts title to take away the focus on the PRO as the manuscript does more deal with the overall outcome of both procedures.

I want to make sure that the term BPO” is properly used in the Introduction session. Please recheck whether the term “BPO” can be used without urodynamic study for each patient.

Answer:

We thank the reviewer for this comment. BPO focuses on the men with diagnosed obstruction without further urodynamic studies, but I agree with you that the term BPE is more appropriate in the manuscript. We have changed the term BPO to BPE throughout the manuscript.

Line 122, What do you mean “intraoperative bleeding”? How did you define and measure?

Answer:

We thank the reviewer for this remark. As this was a retrospective analysis, all surgical reports were analysed as to each surgeon’s remark on extended intraoperative bleeding with the necessity for extensive coagulation compared to a standard case of PVP or TUR-P. We use standard texts for the procedure, therefore an additional remark on intraoperative bleeding has to be actively documented. Therefore, we feel this is valid to be reported. Although bleeding is higher in TUR-P overall the documented intraoperative bleeding in TUR-P was relatively low with 16%. We have added a statement in line 125 to clarify.

In the Table 4, You included bladder neck contracture, erectile dysfunction & retrograde ejaculation in “early adverse events (<30 days: cases)”. But it is difficult to think those events are meaningful complications in the early postoperative period. Rather those should be measured and compared in the late stage. What is your thoughts?

Answer:

We thank the reviewer for this essential remark. The complications are stated according to the time of occurrence as reported by the patients. Therefore, we believe that these complications should still be stated. We have added an explanation in the method section line 75. We have the point "retrograde Ejaculation." taken out because this complication was not explicitly requested, but one patient mentioned it.

In Conclusions session, the authors still summarizes several things rather than PRO itself. So it would be better to be revised little bit.

Answer:

We thank the reviewer for the important remark. We have changed the title for fit the content of the manuscript, therefore we believe the story line of the manuscript can stay unchanged. We hope the reviewer agrees with us.